# Association of Maternal Observation and Motivation (MOM) Program with m-Health Support on Maternal and Newborn Health

**DOI:** 10.3390/healthcare9121629

**Published:** 2021-11-25

**Authors:** Premalatha Paulsamy, Vigneshwaran Easwaran, Rizwan Ashraf, Shadia Hamoud Alshahrani, Krishnaraju Venkatesan, Absar Ahmed Qureshi, Mervat Moustafa Arrab, Kousalya Prabahar, Kalaiselvi Periannan, Rajalakshimi Vasudevan, Geetha Kandasamy, Kumarappan Chidambaram, Ester Mary Pappiya, Kumar Venkatesan, Vani Manoharan

**Affiliations:** 1College of Nursing, Mahalah Branch for Girls, King Khalid University, Khamis Mushaiyt 61421, Saudi Arabia; pponnuthai@kku.edu.sa (P.P.); shalshrani@kku.edu.sa (S.H.A.); marb@kku.edu.sa (M.M.A.); 2Department of Clinical Pharmacy, College of Pharmacy, King Khalid University, Abha 62529, Saudi Arabia; vickku.e@gmail.com (V.E.); glakshmi@kku.edu.sa (G.K.); 3Department of Pharmacology, University College of Medicine and Dentistry, The University of Lahore, Lahore 55150, Pakistan; drriz72@yahoo.com; 4Department of Pharmacology, College of Pharmacy, King Khalid University, Abha 62529, Saudi Arabia; aqureshi@kku.edu.sa (A.A.Q.); raja@kku.edu.sa (R.V.); kumarappan@kku.edu.sa (K.C.); 5Family and Community Health Nursing, Faculty of Nursing, Menoufia University, Shibin el Kom 32511, Egypt; 6Department of Pharmacy Practice, Faculty of Pharmacy, University of Tabuk, Tabuk 71491, Saudi Arabia; kgopal@ut.edu.sa; 7Department of Mental Health Nursing, Oxford School of Nursing & Midwifery, Faculty of Health and Life Sciences, Oxford Brookes University, Oxford OX3 0FL, UK; kperiannan@brookes.ac.uk; 8Regional Nursing Administration, Directorate of General Health Affair, Ministry of Health, Najran 21431, Saudi Arabia; epappiya@moh.gov.sa; 9Department of Pharmaceutical Chemistry, College of Pharmacy, King Khalid University, Abha 62529, Saudi Arabia; kumarve@kku.edu.sa; 10Georgia CTSA, Emory University Hospital, Atlanta, GA 30078, USA; vani.manoharan@emoryheslthcare.org

**Keywords:** pregnant mothers, physical activity, maternal well-being, antenatal mothers, newborn outcomes, m-health, low birth weight, small for gestation, gestation age, hemoglobin

## Abstract

Maternal and child nutrition has been a critical component of health, sustainable development, and progress in low- and middle-income countries (LMIC). While a decrement in maternal mortality is an important indicator, simply surviving pregnancy and childbirth does not imply better maternal health. One of the fundamental obligations of nations under international human rights law is to enable women to endure pregnancy and delivery as an aspect of their enjoyment of reproductive and sexual health and rights and to live a dignified life. The aim of this study was to discover the correlation between the Maternal Observation and Motivation (MOM) program and m-Health support for maternal and newborn health. A comparative study was done among 196 pregnant mothers (study group—94; control group—102 mothers) with not less than 20 weeks of gestation. Maternal outcomes such as Hb and weight gain and newborn results such as birth weight and crown–heel length were obtained at baseline and at 28 and 36 weeks of gestation. Other secondary data collected were abortion, stillbirth, low birth weight, major congenital malformations, twin or triplet pregnancies, physical activity, and maternal well-being. The MOM intervention included initial face-to-face education, three in-person visits, and eight virtual health coaching sessions via WhatsApp. The baseline data on Hb of the mothers show that 31 (32.98%) vs. 27 (28.72%) mothers in the study and control group, respectively, had anemia, which improved to 27.66% and 14.98% among study group mothers at 28 and 36 weeks of gestation (*p* < 0.001). The weight gain (*p* < 0.001), level of physical activity (*p* < 0.001), and maternal well-being (*p* < 0.01) also had significant differences after the intervention. Even after controlling for potentially confounding variables, the maternal food practices regression model revealed that birth weight was directly correlated with the consumption of milk (*p* < 0.001), fruits (*p* < 0.01), and green vegetables (*p* < 0.05). As per the physical activity and maternal well-being regression model, the birth weight and crown–heel length were strongly related with the physical activity and maternal well-being of mothers at 36 weeks of gestation (*p* < 0.05). Combining the MOM intervention with standard antenatal care is a safe and effective way to improve maternal welfare while upholding pregnant mothers’ human rights.

## 1. Background

Pregnancy is a precious event that can influence maternal health and the health of future generations. Nutrition is vital for maternal and child health, and it is well acknowledged that well-balanced nutrition in early life is the foundation for long-term health [1]. Poor maternal nutrition, maternal body composition, metabolism, and placental nutrient supply are key factors that can influence fetal development and have been associated with a poor pregnancy outcomes and fetal genetic potential expression [2].

For maternal, fetal, and newborn health, counseling women to promote awareness of the importance of maternal health prior to conception and in pregnancy and advocating a culturally accepted lifestyle modification in favor of a healthy weight is critical. Although a balanced diet is widely available in today’s globalized world, a shift to a high-fat, low-quality diet has resulted in inadequate vitamin and mineral intake during pregnancy [3]. When the required micronutrient intakes are difficult to meet through food alone, supplementation and fortification can be of benefit [4]. The research does not support routine multiple micronutrient supplementations alone [5]; rather, it emphasizes the significance of an individualized approach to identify nutritional deficiencies in individuals, leading to healthy, culturally accepted lifestyle practices prior to conception and in pregnancy and ultimately to better maternal and newborn outcomes [6].

According to the World Health Organization (WHO) guidelines on “antenatal care (ANC) for a positive pregnancy experience” [1], counseling on healthy eating and staying physically active during pregnancy is recommended for pregnant women who want to stay healthy and avoid accumulating too much weight. Pregnant women should take a daily iron and folic acid supplement with 30 to 60 mg of elemental iron and 400 mcg (0.4 mg) of folic acid to avoid maternal anemia, puerperal sepsis, low birth weight, and premature birth. Each pregnant woman should keep her case notes throughout her pregnancy to ensure continuity, quality of treatment, and a positive pregnancy experience. Antenatal care models with at least eight visits are recommended to reduce perinatal mortality and improve pregnant women’s experiences. It is proposed that maternal and newborn health-related behaviors be promoted to a broad range of cadres, including lay health workers, auxiliary nurses, nurses, midwives, and doctors [1].

“Every mother and newborn receive quality care during the pregnancy, childbirth, and postnatal period”, the WHO envisions [7]. There is evidence that effective intervention strategies for preventing or treating all life-threatening maternal complications are affordable [8]. Optimal adaptation and uptake of existing research findings could alleviate nearly two-thirds of the global maternal and newborn disease burden worldwide [9]. In 2015, almost 303,000 women and adolescent girls died worldwide due to pregnancy and delivery complications [10]. The majority of maternal deaths occur in low-resource settings, and most of them can be avoided [11]; also, almost 2.6 million babies were stillborn in 2015, mostly in low-resource areas [12].

Nevertheless, the association between maternal nutrition and birth outcome is quite complex and influenced by biological, socioeconomic, and demographic factors. Understanding the relationship between maternal nutrition, pregnancy, and birth outcomes may serve as the basis for nutritional interventions. It will improve birth outcomes and the newborn’s long-term health, improving quality of life and lowering mortality, morbidity, and healthcare costs [13]. 

Under the international human rights law, one of the fundamental commitments of nations includes enabling women to survive pregnancy and delivery as part of their enjoyment of sexual and reproductive health and rights and living a life of dignity [14]. The human-rights-based approach promotes health and well-being while preserving dignity and rights and also avoiding substantial morbidity and mortality.

For this, based on a woman’s preferences and available possibilities, we need novel, evidence-based methods of antenatal care to accomplish the “Every Woman Every Child” vision and the Global Strategy for Women’s, Children’s, and Adolescents’ Health. Maternal care also provides an opportunity to interact with and support women, families, and communities at a crucial period in a woman’s life.

### 1.1. Significance of the Study

Poor maternal nutrition, resulting in either malnutrition or excessive weight gain, is common and significant in poor perinatal outcomes [15]. Low BMI in early pregnancy augments the risk of both small for gestational age (SGA) babies and preterm birth [16]. A study showed that high BMI increased the risk of preterm birth and also low weight gain, which is associated with SGA and preterm birth [17]. According to randomized control trials (RCTs), it is well proven that intervention improves birth outcomes, especially when given to impoverished women leads to their newborns gain roughly an extra 100 g (95% CI: 56–145) at delivery [18]. The WHO has set a global target to reduce LBW by 30% by 2025 [19].

Iron, folate, and vitamin A deficiency are all linked to anemia and associated with an increased risk of maternal and infant mortality [20]. Anemia also significantly increases the danger of maternal death by a factor of two [21]. In 2017, more than 50 million children had BMIs that were too low as a result of inadequate nutrition during pregnancy [22]. In addition to producing anemia, iron deficiency harms muscular energy usage and, as a result, physical capability and work performance. It also affects the immunological state and, thereby, infection-related morbidity [23]. Folate (vitamin B9) insufficiency has been related to prenatal neural tube abnormalities in addition to anemia [24,25]. Iron–folic acid supplementation substantially decreased self-reported intra-partum hemorrhage in an RCT in rural Nepal [26].

Pre-eclampsia has been related to calcium deficiency [27], and shortages of other vitamins and minerals, such as vitamin E, C, B6, and zinc, have also been suggested to have a role in the condition [28]. Zinc insufficiency is linked to weakened immunity [29,30,31]. Due to limited access to comprehensive maternal healthcare, women from low-income, rural, and other marginalized backgrounds are more likely to experience pregnancy- and childbirth-related complications [32].

Pregnancy is a time of physical and emotional stress, and psychological symptoms occur at a higher rate in pregnant women, adversely affecting pregnancy and postpartum outcomes [33]. It is, therefore, necessary to routinely screen for psychological symptoms during pregnancy and postpartum.

Mobile health, or “m-Health”, technologies are increasingly being used to improve chronic disease management and encourage behavior change [34]. Given that more pregnant women are using smart phones, mobile health provides an opportunity to improve pregnancy health behaviors. Hence, this study was undertaken to find the impact of the Maternal Observation and Motivation (MOM) program with m-Health support on maternal and newborn health.

### 1.2. Objectives

To test the hypothesis that the Maternal Observation and Motivation (MOM) program with m-Health support during pregnancy would improve maternal and newborn health.

## 2. Methods

### 2.1. Design

A comparative study was done to assess the effect of the Maternal Observation and Motivation (MOM) program with m-Health support on maternal and newborn health. 

### 2.2. Participants and Setting

The study participants were pregnant women with gestational age not less than 20 weeks with a viable fetus confirmed by ultrasonogram. The study took place in six villages, 30–40 km from Chennai city and covered a population of 35,000. The eligible samples were 196 pregnant women who met the inclusion criteria and were ready to take part in the study. 

### 2.3. Sample Size and Sampling Process

The non-probability convenient sampling technique selected 196 samples from six villages in the Chennai suburban area. The inclusion criteria were viable fetus, gestational age not less than 20 weeks, no severe illness, written consent for participation, and possession of a smart phone when recruited. The mothers from three villages were allotted to the study group (94 mothers) and those from three other villages were in the control group (102 mothers) for convenience and to avoid contamination. The CONSORT diagram is included in Figure 1.

### 2.4. Data Collection Tools/Instruments

Section A: The sociodemographic characteristics of the participants included age, parity, education, occupation of the mothers, husband’s education and occupation.

Section B: Food Frequency Questionnaire (FFQ) was created and used to collect the baseline data on diet history. 

Section C: Maternal outcomes such as height (Ht), weight (kg), hody mass index (BMI), and hemoglobin (Hb) at baseline, 28 weeks, and 36 weeks were obtained. In addition, the data related to maternal morbidity and mortality, such as spontaneous abortion, induced abortion, stillbirth, or live births, were obtained from a structured checklist. 

Section D: Newborn outcomes such as birth weight and crown–heel length were measured to the nearest 0.1 cm using fiberglass tapes (CMS Instruments, London, UK). Secondary data such as low birth weight, small for gestational age, major congenital malformations, and twin or triplet pregnancies were also collected. 

Section E: The data were gathered using a self-structured pregnancy physical activity (PA) checklist. 

Section F: Maternal well-being was assessed using the World Health Organization-5 Well-Being Index (WHO-5) [35] tool.

### 2.5. Assessment

The participating pregnant women were followed up monthly at home by one of the investigators or research assistants. Information on specified gastrointestinal adverse effects was collected after four weeks of supplementation and the MOM program. If the mother had minor ailments such as constipation or heartburn, home remedies as per the WHO guidelines were given. If those symptoms persisted or worsened, mothers were referred to the nearby health centers/hospitals. The weight and Hb level in venous blood were assessed using HemoCue (HemoCue AB) at initial contact, 28-, and 36-weeks of gestation. Anemia was defined as having a hemoglobin level less than 11.0 g/mL. All participants in both groups were provided with two 3-day food diaries to quantify intakes and physical activity surveys to ascertain self-reported compliance, monitored by the investigating team.

Pregnancy outcomes were classified as spontaneous abortion (unintended loss of a fetus before 28 weeks’ gestation as determined by reported last menstrual period), induced abortion (intentional loss of a fetus before 28 weeks’ gestation), stillbirth (birth of a dead fetus after 28 weeks’ gestation), or live birth (birth of a fetus with any sign of viability). The birth anthropometry was primarily measured by trained nurses within 72 h of birth. SECA electronic scales (SECA GmbH) were used to measure all birth weights to a precision of 10 g.

The crown–heel length of the newborn was measured using standard procedures using domestically manufactured collapsible length boards that were accurate to 1 mm. Training of the interviewers/research assistants on anthropometric measurements was conducted periodically. Weighing equipment was calibrated monthly with standard weights. Inter-observer and intra-observer variation studies were conducted every month to ensure the quality of these measurements. The data were gathered using a self-structured pregnancy physical activity (PA) checklist. 

The duration of each day’s physical activity was recorded in the self-monitored diary. The “Fit” app was downloaded on the mothers’ smart phones to observe and motivate their PA to calculate the average weekly minutes spent on physical activity. Pregnancy is a time of physical and emotional stress, as well as progressive physiological and psychological changes. Psychological symptoms occur at a higher rate in pregnant women than in non-pregnant women and adversely affect pregnancy and postpartum outcomes. Therefore, the WHO-5, a short-form (five items) tool was used. The five items are as follows: I have felt cheerful and in good spirits; I have felt calm and relaxed; I have felt active and vigorous; I woke up feeling fresh and rested; and my daily life has been filled with things that interest me. Each item is rated on a six-point Likert-type scale of 0–5, with 0 indicating the lack of positive feelings during the preceding 2 weeks and 5 indicating consistent positive feelings. The total raw score, which ranges from 0 to 25, is multiplied by 4 to obtain the final score, with 0 representing the worst possible well-being and 100 representing the best possible well-being [26].

### 2.6. Intervention

Women in the control group received only the standard antenatal care, and the intervention group received routine antenatal care as well as a the “Maternal Observation and Motivation (MOM)” program. The MOM program began with a single face-to-face education session conducted for a group of 5–8 antenatal mothers.

This education session was delivered at the first visit and emphasized the importance of nutrition; the effect of anemia and vitamin or mineral deficiency on the mother, the fetus, and the child in later life; the compliance of supplemental medications and nutrition supplement mixes; and the importance of the physical activity and emotional well-being of the mother during pregnancy. Participants were encouraged and were informed about a healthy diet, including adding fresh fruits and vegetables, milk and milk products. The recommended diet provided additional calories, proteins, and micronutrients that approximated the recommended levels. 

Under the Integrated Child Development Services (ICDS) program [36], the antenatal (AN) mother receives a supplemental nutrition powder that has the following components in 100 mg: protein (11 g), carbohydrate (55 g), calcium (200 mg), vitamin A (200 mcg), niacin (4 mg), folic acid (15 mcg), fat (3 mg), iron (6 mg), vitamin B1 (300 mcg), vitamin B2 (350 mcg), vitamin C (15 mg), and calories (400 kcals). The AN mother receives 165 g/day as 15 days ration from Anganwadi centers. In addition to the nutritional mix given by the Government of Tamilnadu, all AN women with Hb = 11 g/dL were given 100 tablets of iron (100 mg) and folic acid (0.5 mg) (1 tablet/day) at 18 week gestation or soon after being registered in the study through the mother and child protection card (MCP Card), according to the National Nutritional Anemia Control Program. If the AN woman had an Hb of 9–10.9 g/dL, 2 tablets/day for 100 days were provided, and if Hb was 7.1 to 8.9 g/dL an additional two tablets per day were given and the AN mother was given an iron sucrose injection. Before starting these iron and folic acid supplements, the AN woman was given albendazole 400 mg tablets for anthelmintics treatment. The same protocol was followed in this study too. It was ensured that the control group mothers also followed the same protocol for ethical reasons.

All women in the study group were given 100 tablets of iron (60 mg) and folic acid (0.5 mg) at initial contact according to the National Nutritional Anemia Control Program and the nutritional mix given by the government Tamilnadu 36]. Anemic mothers were given special consideration and followed the above protocol. The information received at the face-to-face education session was reinforced by smart phone, through WhatsApp messages every 2 weeks (sent by the research team). The research assistants did three follow-up, face-to-face, in-person home visits besides those at baseline, 26 weeks, 36 weeks of gestation, and after delivery. Among the four research assistants, three were nurses with master’s degrees in Obstetrics and Gynecology and one had a masters in Mental Health Nursing. The research assistants were given training on the assessment of data and counseling of mothers related to MOM and motivating the mothers to comply with the nutrition habits, supplements, regular physical activity, and mental well-being.

The content of the WhatsApp messages was standardized to a specific theme on a 2 week basis with some discourse between the researchers and participants. Individual attention was given to each mother on her concerns about diet, physiological alterations related to pregnancy and its’ management, and follow up of nutrition supplementation. Individualized counseling was also given on minor ailments related to pregnancy, and one of the researchers also visited the mothers at least two times during the study period. The virtual meetings occurred eight times during the study period with three in-person visits.

### 2.7. Ethical Consideration

Official permission to conduct the study and ethical approval were obtained from the Institutional Ethical Committee with ICE/LCN/2021-02 dated 29 January 2021. Consent from the participants was collected before starting the study after explaining the aim of the study, their role, the confidentiality of the information, and their right to depart from the study at any point of data collection. A no-harm certificate was obtained from an obstetrician for the MOM program. The control group mothers were also assured that they were following the standard antenatal care protocol of the state government. Confidentiality and beneficence were assured throughout the study period.

### 2.8. Statistical Analysis

The data were processed and analyzed by SPSS version 21.0. Armonk, NY using descriptive and inferential statistics. Analysis of data was done in accordance with intention to treat. Analysis of variance was used to examine the main effects of food practices among the two groups on the Hb level of the mothers. The ‘*t*’ test was used to compare differences between group means to compare the MOM intervention with the standard program. Multiple regression analysis was used to examine changes in birth weight of the newborn according to maternal food practices, physical activity, and maternal well-being. In regressions, birth measurements and maternal measurements were evaluated as continuous variables. Food practices, physical activity, and maternal well-being scores were analyzed as grouped variables, and *p* < 0.05 was considered statistically significant.

## 3. Results

Table 1 shows the baseline data of the mothers of the study and control group. In this study, 94 mothers and 102 mothers in the study group and in the control group respectively. The mean age of the mothers was 28.1 ± 4.6 and 27.9 ± 4.8 years, respectively, for the study and control groups. Regarding the parity, 48.9% and 53.92% were primipara mothers in the study and control group. The mean height of the mothers was 144.8 cm with an SD of 6.1 in the study group, whereas it was 147 ± 6.9 cm in the control group. The average BMI was 21.7 ± 3.1 and 21.3 ± 2.9 for the study and control group, respectively. The majority of the mothers had education up to secondary school (65.96% vs. 55.88%) and most of them were not working (77.66% vs. 69.71%). The education status of the husbands was more or less similar in both groups (73.4% vs. 76.47%), and a similar trend was also seen in their occupation status.

The dietary intake of the pregnant mothers was measured using a self-reported FFQ checklist on milk and milk products, green leafy vegetables, and fruits. These data were used for the regression analysis to determine their relationship with the maternal and newborn outcomes. In addition to this, the mothers were given nutritional supplements as per the Government of Tamilnadu.

Table 2 shows the comparison of Hb status among study and control group mothers at different periods of observation. For the study group, the mean Hb was 110.4 with an SD of 3.08 at baseline, 112.7 with an SD of 4.13, and 113.1 with SD of 3.16 at 28 and 36 weeks of gestation at *p* < 0.001 as per repeated ANOVA.

The baseline data on Hb show that 31 (32.98%) vs. 27 (28.72%) of the study and control group had anemia, which was improved to 27.7% and 15.0% among study group mothers at 28 and 36 weeks of gestation. There was a significant dissimilarity in anemia status among the study and control group at the *p* < 0.001 level after intervention. The weight gain among the intervention group also showed a significant disparity at the *p* < 0.001 level. There was a significant variation between physical activity and maternal wellbeing groups at the *p* < 0.001 and *p* < 0.01 levels (Table 3). Similarly, the crown–heel length was 47.6 ± 3.0 vs. 46.8 ± 3.5 cm among the study and control group, respectively.

The newborn outcomes show a significant difference between groups at *p* < 0.001 in crown–heel length and in birth weight and gestational age at birth at *p* < 0.01. The mean birth weight of the study group was slightly higher than that of the control group (2.78 ± 0.56 vs. 2.56 ± 0.49 kg). Similarly, the crown–heel length was 47.6 ± 3.0 vs. 46.8 ± 3.5 cm in the study and control group, respectively.

The birth weight was strongly related with the consumption at 36 weeks of gestation of milk (*p* < 0.001), fruits (*p* < 0.01), and green vegetables (GLV) (*p* < 0.05) even after adjustment for potentially confounding variables such as sex, parity, gestational age at birth, and baseline weight of mothers (Table 4).

The newborn’s birth weight and crown–heel length was strongly connected with the physical activity at 36 weeks of gestation (*p* < 0.001) after adjusting the predictor variables such as sex; parity; and baseline weight of the mother; and intake of milk products, GLV, and fruits. Maternal well-being also showed a significant influence on the birth weight and crown–heel length of the newborn (*p* < 0.05) (Table 5).

The data on preterm birth (PTM), spontaneous abortion, stillbirth, low birth weight, small for gestational age, major congenital malformations, and twin or triplet pregnancies were obtained from a structured checklist. There were only two spontaneous abortions and four small-for-gestation babies among the study participants (Figure 2).

## 4. Discussion

This study was done to compare the effect of the Maternal Observation and Motivation (MOM) program on maternal outcomes such as weight (kg) and Hb at baseline, 28 weeks, and 36 weeks and newborn outcomes such as birth weight, crown–heel length, and gestational age at birth. The data on preterm birth (PTM), spontaneous abortion, stillbirth, low birth weight, SGA, major congenital malformations, and twin or triplet pregnancies were obtained from a structured checklist. The strengths of our study were that it was community based, and the mothers were continuously observed and motivated by the qualified nurses for their compliance with the intervention as well as to promote their well-being, throughout their pregnancy. In addition, according to the WHO recommendations that pregnant women be followed up at least eight times during their pregnancy, in this project, the mothers had four face-to-face counseling sessions and eight virtual follow-ups with the healthcare professionals through smart phones.

The baseline data on Hb of the mothers show that around 29% of them had anemia, which was improved to 27.7% and 15.0% among study group mothers at 28 and 36 weeks of gestation after the intervention. According to the findings of a study, globally, in 2016, an estimated 35.3 million pregnant women worldwide or 40% were anemic. Among them, 19% of women are affected by iron deficiency [37]. The WHO also says that anemia among pregnant women ranges from 14% in affluent countries to 65–75% in India [38]. Even though anemia is easily curable and avoidable illness, it is strongly linked to pregnancy. Increasing each 1 g/dL of mean hemoglobin (Hb) in late pregnancy reduces the odds of maternal deaths by 29% [39]. Reduced intake, excess demand in multigravida women, altered metabolism, low socioeconomic status, illiteracy, early marriage, and infectious diseases such as hookworm infestations may be the causal factors linked to the occurrence of anemia during pregnancy [40]. Hence, the improved Hb among mothers of this study caused by the MOM intervention enhanced the pregnancy-related outcomes. Thus, nutrition education and counseling during pregnancy is a promising strategy to improve maternal and newborn health.

In addition, inadequate folate levels in the blood before conception and in the first trimester can cause severe fatal neural tube abnormalities. Spina bifida and anencephaly, the two most prevalent kinds of neural tube disorders, affect an estimated 300,000 babies each year [38,41]. Therefore, it is crucial to improve pregnant mothers’ Hb and folic acid level to elevate the health status of the fetus and future generations.

According to a few prospective studies on maternal nutrition, after controlling for a variety of covariates, the maternal thinness, as defined by a low mid-upper arm circumference, was linked to all-cause maternal death (up to 42 days postpartum) in Nepal [26] as well as severe hemorrhage and sepsis morbidity in Bangladesh [42]. The study on “Improving Women’s Diet Quality Preconceptionally and during Gestation: Effects on Birth Weight and Prevalence of Low Birth Weight” found that there was a significant disparity in anemia status and weight gain among the study and control groups at *p* < 0.001 after the intervention, which denied the effect of daily snacking with leafy vegetables, fruit, and milk before conception and during pregnancy on the birth weight [43]. Hence, the tailor-made health coach approach of this present study, with the additional emphasis on physical activity and mothers’ mental health proved to be effective. There were only two spontaneous abortions and four small-for-gestation babies among the study participants (Figure 2).

As per the concept, the WHO’s shift of considering diet quality and promoting healthy eating practices on antenatal care recommends counseling on healthy eating and keeping physically active during pregnancy [44]. Furthermore, under the international human rights law, one of the fundamental commitments of nations include enabling women to survive pregnancy and delivery, as part of their enjoyment of sexual and reproductive health rights, as well as living a life of dignity [14]. Hence, by considering the above facts and providing a nutritious diet, pregnant women need additional support in the form of emotional and social motivation and individualized care during pregnancy.

This study’s strong point was that it adopted an individualized approach for pregnant women with in-person interactions and the most novel approach of m-Health support through smart phones. This allowed the investigators to meet the mothers often and give custom-made interactions and resolutions for the women. This was reflected in the study findings on physical activity, maternal well-being, and the outcome variables of the study. There was a significant dissimilarity between physical activity and maternal well-being among groups in the present study at *p* < 0.001 and *p* < 0.01 levels.

Nearly three million annual newborn deaths account for an estimated 44% of all under-five deaths globally. There is solid evidence for various therapies on prevention of this, provided alone or in packages, across both time and place continuums (preconception, pregnancy, birth, and postnatal). While these interventions can prevent 70% of newborn fatalities (and more than one-third of stillbirths) [45], the most challenging issues ahead are overcoming bottlenecks and healthcare system barriers to achieve high coverage and quality at scale. In the present study, with the MOM intervention, the newborn outcomes show significant differences at *p* < 0.001 in crown–heel length, birth weight, and gestational age at birth at *p* < 0.01, respectively. The mean birth weight of the study group was slightly higher than that of the control group (2.780 ± 0.560 vs. 2.560 ± 0.490 kg). Similarly, the crown–heel length was 47.6 ± 3.0 vs. 46.8 ± 3.5 cm among the study and control groups. This could have been because the present study provided individual attention to the antenatal mothers and included regular monitoring and motivation to improve the mothers’ physical and mental well-being of the mothers.

According to the maternal food practices regression model, the birth weight was strongly associated with the consumption at 36 weeks of gestation of milk (*p* < 0.001), fruits (*p* < 0.01), and green vegetables (*p* < 0.05) even after adjustment for potentially confounding variables such as gender, parity, gestational age at birth, and baseline weight (Table 4). This gives the insight that consuming the supplements alone will not be enough to improve the birth weight. Consumption of milk and milk products, fresh fruits, vegetables, and green leafy vegetables play a significant role other than nutrition supplementation on the fetal growth and birth weight.

As per the physical activity and maternal well-being regression model, the birth weight and crown–heel length were strongly related with the physical activity and maternal well-being of mothers at 36 weeks of gestation (*p* < 0.05) even after adjustment for potentially confounding variables such as gender; parity; gestational age at birth; and baseline intake of milk products, GLV, and fruits. This finding suggests that the newborn outcome variables such as birth weight and crown–heel length were strongly linked with the mothers’ physical activity and maternal well-being of the mothers.

Hence, this multi-component MOM intervention incorporating diet, nutritional supplements, physical activity, and maternal well-being by personal interaction with continuous observation and motivation through m-Health improved the maternal and newborn outcomes significantly. However, among the independent variables, physical activity and maternal well-being played a significant role in the improvement of newborn outcomes in addition to dietary habits, including intake of milk products, GLV, and fruits.

Our results contrast with the results of RCTs conducted by two prospective studies [38,43] that meagre nutrition supplementation and dietary modification were unsuccessful in bringing the desired changes in the newborn outcome variables such as birth weight, length, and gestational age at birth. Since optimal maternal weight and Hb are the essential components of maternal and newborn care, improving these outcomes by modifying dietary habits and improving physical activity and maternal well-being by continuous monitoring with motivation is sensible for any intervention trials. The interventions including incessant motivation may enhance the compliance of nutrition supplements, dietary modification, adherence to expected physical activity, and improved maternal well-being, which might have improved the maternal and newborn outcomes in study participants.

## 5. Conclusions

While reducing maternal mortality is an important indicator, simply surviving pregnancy and childbirth does not imply improved maternal health. The burden of maternal morbidity can have long-term consequences for women’s health and well-being. Adopting a human rights framework for universal health necessitates the provision of high-quality care not only during pregnancy and labor but also before and after childbirth. It is critical to broaden the focus on mortality to include morbidity to achieve health for all. This MOM intervention involving face-to-face counseling, constant motivation, and observation by m-Health improved compliance, physical activity behavior, and maternal well-being and eventually improved maternal and newborn health. Hence, it is a safe and effective measure to use the MOM intervention with standard antenatal care to improve maternal well-being from the human rights point of view and reduce the disease burden.

## Figures and Tables

**Figure 1 healthcare-09-01629-f001:**
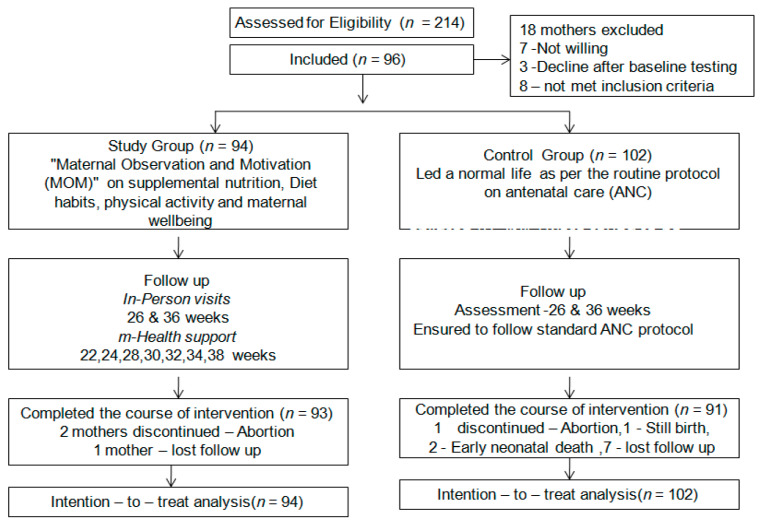
CONSORT flow diagram of the study.

**Figure 2 healthcare-09-01629-f002:**
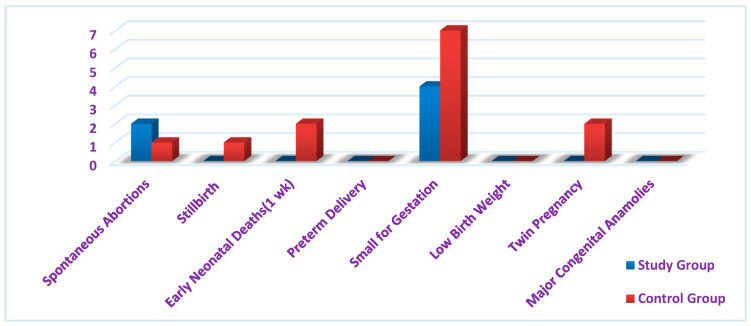
Distribution of secondary variables.

**Table 1 healthcare-09-01629-t001:** Baseline characteristics of the pregnant women of both the study and the control group.

Variables	Study (*n* = 94) (%)	Control (*n* = 102) (%)
Age (in years) Mean ± SD	28.14.6	27.9 ± 4.8
Parity		
Primipara mothers	46 (48.9)	55 (53.9)
Multipara mothers	48 (51.1)	47 (46.1)
Height (cm)	144.8 ± 6.1	147 ± 6.9
Weight (kg)	45.8 ± 4.3	46.1 ± 3.9
BMI (kg/m^2^)	21.7 ± 3.1	21.3 ± 2.9
Educational Attainment		
Primary or less	21 (19.7)	19 (18.6)
Up to secondary	62 (66.0)	57 (55.9)
Completed college education	11 (11.7)	26 (25.5)
Occupation		
Semiskilled/unskilled	16 (17.0)	19 (18.6)
Skilled/self-employed	3 (3.2)	5 (4.9)
Professional	2 (2.1)	7 (6.7)
Not working	73 (77.7)	71 (69.1)
Husband’s Education		
Primary or less	17 (18.7)	16 (15.7)
Secondary	69 (73.4)	78 (76.5)
Graduate	8 (8.5)	8 (7.8)
Husband’s Occupation		
Semi-skilled/unskilled	58 (61.7)	62 (60.8)
Skilled/self-employed	24 (25.5)	27 (26.5)
Professional	4 (4.3)	6 (5.9)
Not working/other	8 (8.5)	7 (6.8)
Dietary Intake		
Milk and Milk Products (other than in coffee/tea)		
1 time/week	43 (45.7)	50 (49.0)
1–6 times/week	31 (33)	37 (36.3)
≥7 times/week	20 (21.3)	15 (14.7)
Green Leafy Vegetables (GLV)		
1 time/week	23 (24.5)	27 (26.5)
1–6 times/week	61 (64.9)	58 (56.9)
≥7 times/week	10 (10.6)	17 (16.7)
Fruits		
1 time/week	14 (14.9)	19 (18.3)
1–6 times/week	59 (55.5)	62 (60.8)
≥7 times/week	21 (22.3)	21 (20.6)

**Table 2 healthcare-09-01629-t002:** Comparison of Hb status among study and control group mothers at different periods of observation.

Maternal Hb Level	Study Group Mean (SD)	Control Group Mean (SD)	*t* Value *p*-Value
Hb level at baseline, g/L	110.4 (3.08)	110.1 (2.91)	*t* = 0.848 *p* = 0.397 (N.S.)
Hb level at 28 week gestation, g/L	112.7 (4.13)	112.1 (3.99)	*t* = 0.199 *p* = 0.843 (N.S.)
Hb level at 36 week 113.1 gestation, g/L (3.16)		112.5 (3.05)	*t* = 0.985 *p* = 0.21 (N.S.)
Repeated ANOVA F = 19.305, *p* < 0.001		F = 0.633, *p* = 0.532	

**Table 3 healthcare-09-01629-t003:** The maternal and newborn outcomes among pregnant mothers.

Maternal Outcome Variables	Study Group (*n* = 94)	Control Group (*n* = 102)	*p*
Anemia (Hb < 110 g/L)			
At 28 weeks	26 (27.7%)	31 (30.4%)	0.001
At 36 weeks	14 (15.0%)	19 (18.6%)	
Weight Gain			
At 28 weeks	43.9 ± 6.2	41.8 ± 8.1	
At 36 weeks	47.6 ± 7.4	46.2 ± 6.7	0.001
Physical Activity (min/week)			
Pre-intervention	81 ± 11	86 ± 16	
Post-intervention	131 ± 29	103 ± 17	0.001
Maternal Well-being			
Pre-intervention	54 ± 13	59 ± 18	0.01
Post-intervention	81 ± 19	67 ± 11	
Newborn Outcome Variables			
Birth weight (kg)	2.78 ± 0.56	2.56 ± 0.49	0.01
Crown–heel length (cm)	47.6 ± 3.0	46.8 ± 3.5	0.001
Gestational age at birth (wk)	38.9 (38.6–39.1)	38.6 (38.4–39.0)	0.01

**Table 4 healthcare-09-01629-t004:** Multiple regression analysis of the relationship of the maternal food practices in 36 weeks of gestation with newborn’s birth weight.

Dependent Variable	Independent Variables	Milk Product Intake			Green Leafy Vegetables Intake			Fruit Intake		
		R^2^	β	*P*	R^2^	β	*p*	R^2^	β	*p*
Birth weight of newborn (g)	Sex, parity, gestational age at birth	19.8 (0.87)	16.9	<0.001	23.1 (1.63)	6.7	<0.05	22.1 (0.98)	5.8	<0.01
	Sex, parity, gestational age at birth and baseline weight of mother	29.3 (1.8)	17.5	<0.001	23.4 (0.47)	5.1	<0.05	31.6 (0.83)	9.1	<0.05

**Table 5 healthcare-09-01629-t005:** Relationship between physical activity and maternal well-being at 36 weeks of gestation with newborn outcomes.

Dependent Variable	Independent Variable	Physical Activity	Maternal Well-being
Birth Weight	P1 (Sex, parity, and baseline weight of mother)	<0.05	<0.05
	P1 and intake of milk products, GLV, and fruits	<0.001	<0.05
Crown–Heel Length	P1 (Sex, parity, and baseline weight of mother)	<0.01	<0.05
	P1 and intake of milk products, GLV, and fruits	<0.05	<0.05

## Data Availability

The data presented in this study are available on request from the corresponding author.

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
