# Peer review of "Association of Maternal Observation and Motivation (MOM) Program with m-Health Support on Maternal and Newborn Health"

_healthcare, 2021, doi:10.3390/healthcare9121629_

Round 1
Reviewer 1 Report
The MOM intervention involving face to face counselling, constant motivation and observa-tion by m-Health have improved compliance, physical activity behaviour and maternal wellbeing and eventually improved maternal and newborn health. It is an interesting and important research. But there are some aspects need be improved.
1.Abstract
Line45: 109?------196?
Line56-57: P 0.01----p<0.01?
P 0.05---p<0.05?
2.Background
It was Long and unfocused. Advice to rewrite it.
3.Methods
Line 245: 2.6. Intervention please describe itclearly.
4.Results
Table format should be modified.
Whether the number of decimal digits after the decimal point should maintain uniformity as possible?
5.Writing should be improved further.
Author Response
The MOM intervention involving face to face counselling, constant motivation and observa-tion by m-Health have improved compliance, physical activity behaviour and maternal wellbeing and eventually improved maternal and newborn health. It is an interesting and important research. But there are some aspects need be improved.
- Abstract
Line45: 109?------196?
Line 56-57: P 0.01----p<0.01? ; P 0.05---p<0.05?
Modified
2.Background
It was Long and unfocused. Advice to rewrite it.
Modified
3.Methods
Line 245: 2.6. Intervention please describe it clearly.
Modified
4.Results
Table format should be modified.
Whether the number of decimal digits after the decimal point should maintain uniformity as possible?
Modified
5.Writing should be improved further.
Modified
Reviewer 2 Report
Please view the PDF for my comments and suggestions for edits.

Author Response
Please view the PDF for my comments and suggestions for edits.
The suggestions given by the reviewer was very well appreciated and indeed given, lot of focus on the topic studied.
Modified as per the suggestions given.
Few points to clarify which were highlighted in the pdf by the reviewer:
- Is this the most appropriate sample to do the study since they are given supplements?
Yes, because as per the ref: 1,7, 32 & 43, supplementation alone is not enough. Hence, this study was undertaken in this sample.
- You make grand conclusion that food causing positive effects – How do you know its not the supplements.
The results concludes that the MOM intervention involving face to face counseling, constant motivation and observation by m-Health have improved compliance, physical activity behaviour and maternal wellbeing and eventually improved maternal and newborn health. Also, the control group also, received same supplements as study group, but, the study group showed significant improvements in maternal and newborn outcomes.
- The WHO wellbeing scale:
It is a standard tool by WHO, which has given score interpretation in such a way that the final score to be multiplied by 4; The total raw score, ranging from 0 to 25, is multiplied by 4 to give the final score, with 0 representing the worst imaginable well-being and 100 representing the best imaginable well-being.
ref. https://www.corc.uk.net/outcome-experience-measures/the-world-health-organisation-five-well-being-index-who-5/
- Other modifications were done in the revised manuscript.
Round 2
Reviewer 1 Report
The manuscript was revised well. But does the table format meet the requirements of three-line table?
Author Response
The manuscript was revised well. But does the table format meet the requirements of three-line table?
MODIFIED
The reviewer's suggestions were well taken and I learnt a lot from the review comments.
Reviewer 2 Report
See comment about p-values.

Author Response
MODIFIED THE 'P' VALUE
The reviewer's suggestions were well taken and I learnt a lot from the review comments.